# Identification and Evaluation of Hub Long Noncoding RNAs and mRNAs in High Fat Diet Induced Liver Steatosis

**DOI:** 10.3390/nu15040948

**Published:** 2023-02-14

**Authors:** Jing Sui, Da Pan, Junhui Yu, Ying Wang, Guiju Sun, Hui Xia

**Affiliations:** 1Research Institute for Environment and Health, School of Emergency Management, Nanjing University of Information Science and Technology, Nanjing 210044, China; 2Key Laboratory of Environmental Medicine Engineering, Ministry of Education, School of Public Health, Southeast University, Nanjing 210009, China

**Keywords:** nonalcoholic fatty liver disease, weighted gene coexpression network analysis, lncRNA, lycopene

## Abstract

Nonalcoholic fatty liver disease (NAFLD) is considered the most prevalent chronic liver disease, but the understanding of the mechanism of NAFLD is still limited. The aim of our study was to explore hub lncRNAs and mRNAs and pathological processes in high-fat diet (HFD)-induced and lycopene-intervened liver steatosis. We analyzed the gene profiles in the GSE146627 dataset from the Gene Expression Omnibus (GEO) database to identify differentially expressed lncRNAs and mRNAs, and we constructed coexpression networks based on weighted gene coexpression network analysis (WGCNA). The Gene Ontology (GO) and Kyoto Encyclopedia of Genes and Genomes (KEGG) databases were utilized for functional enrichment analysis. We found that the turquoise, blue, brown, yellow, green, and black modules were significantly correlated with NAFLD. Functional enrichment analysis revealed that some hub lncRNAs (Smarca2, Tacc1, Flywch1, and Mef2c) might be involved in the regulation of the inflammatory and metabolic pathways (such as TNF signaling, metabolic, mTOR signaling, MAPK signaling, and p53 signaling pathways) in NAFLD. The establishment of an NAFLD mouse model confirmed that lycopene supply attenuated hepatic steatosis in HFD-induced NAFLD. Our analysis revealed that the inflammatory and metabolic pathways may be crucially involved in the pathogenesis of NAFLD, and hub lncRNAs provide novel biomarkers, therapeutic ideas, and targets for NAFLD. Moreover, lycopene has the potential to be a phytochemical for the prevention of HFD-induced liver steatosis.

## 1. Introduction

Nonalcoholic fatty liver disease (NAFLD) is considered the most prevalent chronic liver disease and is one of the leading causes of death worldwide, with the number of cases increasing [1,2]. Hepatic parenchymal steatosis is the hallmark of NAFLD, without alcohol consumption or other clear factors for liver damage. Moreover, the development of NAFLD is related to obesity, insulin resistance, type 2 diabetes, hyperlipidemia, and metabolic syndrome [3].

Lycopene is a carotenoid widely found in red, pink, and orange fruits and vegetables, such as tomatoes, papaya, and grapes [4]. Lycopene is well-known for its antioxidant and anti-inflammatory activities, as well as regulating various vital metabolic pathways [5,6,7]. Lycopene functions as a free radical scavenger, so can alleviate lipid peroxidation damage and protect the liver [8]. Studies have reported that lycopene can activate the essential signaling pathway in redox homeostasis (nuclear factor erythroid 2-related factor 2 pathway) to promote the expression of antioxidant genes, reduce oxidative stress, and repair liver damage [9,10]. Additionally, several preclinical and clinical studies have been conducted to validate the protective and preventive effects of lycopene on NAFLD [11].

Increasing numbers of studies are revealing that long noncoding RNAs (lncRNAs) play important biological roles in various types of diseases [12,13], including NAFLD [14,15]. In the cytoplasm, lncRNAs can bind to messenger RNAs (mRNAs) to affect the splicing, maturation, transport, stability, and translation efficiency of mRNAs [16]. Accumulating evidence suggests that lncRNAs play important regulatory roles in the pathophysiology and organ function of NAFLD [17]. For example, lncRNA Blnc1 plays a regulatory role in inducing NAFLD via the activation of the LXR-SREBP1c pathway [14]. lncRNA H19 interacts with polypyrimidine-tract-binding protein 1 (PTPB1) to regulate hepatic lipid metabolism in NAFLD [18]. Studies have shown that predicting the functions of lncRNAs based on the coexpression network of lncRNAs and mRNAs is beneficial for further research on NAFLD [19,20,21].

Weighted gene coexpression network analysis (WGCNA) is a new algorithm for analyzing modules from gene expression profiling data [22]. WGCNA describes how genes (including hub lncRNAs and mRNAs) interactively work and identifies the correlation between highly coexpressed modules. However, current studies have focused on the interaction between mRNAs with WGCNA in NAFLD [23,24,25], whereas no study has performed WGCNA to construct lncRNA–mRNA coexpression networks concerning NAFLD. To explore hub lncRNAs and mRNAs and the pathological processes in high-fat diet (HFD)-induced and lycopene-intervened liver steatosis, we identified lncRNAs and mRNAs based on RNA sequencing data (GSE146627) from the Gene Expression Omnibus (GEO) database. WGCNA was used to mine lncRNA and mRNA modules and predict the targeted relationships between lncRNAs and mRNAs. After constructing the lncRNA–mRNA network, we explored the regulation of mRNA by lncRNA in specific pathways, which provides a better understanding of the biological function of lncRNA.

## 2. Materials and Methods

### 2.1. Data Retrieval and Processing

The expression profiling of GSE146627, which is based on the GPL21273 platform (Illumina HiSeq X Ten (Mus musculus), Vazyme Biotech Co., Ltd, Nanjing, Jiangsu, China), was downloaded from the GEO database (https://www.ncbi.nlm.nih.gov/geo/query/acc.cgi?acc=GSE146627 (accessed on 20 April 2022)). GSE146627 contained nine liver samples: three fed a normal diet, three fed a HFD, and three fed HFD mixed with lycopene. The fastq raw data were subjected to mapping with grcm38 reference genomes by using the HISAT2 and StringTie packages [26]. DEseq2 in R was used to screen differentially expressed lncRNAs and mRNAs among the three groups of samples [27]. The thresholds were set to *p* < 0.05, fold change > 2, and false discovery rate < 0.05. The workflow of overall study design is shown in Figure 1.

### 2.2. Weighted Gene Coexpression Network Analysis (WGCNA)

WGCNA is a method used for finding highly synergistic expressed gene modules and the association between gene sets and disease [28]. WGCNA was performed using the R package “WGCNA” on all genes. To maximize the integrity of the statistical information, 1602 lncRNAs and 2202 mRNAs were selected for WGCNA with the “WGCNA” package in R software (Version 4.1.1). We further determined whether two genes had similar expression patterns by calculating the Pearson coefficient between any two genes. The β value indicates a soft threshold power of the independence and the average connectivity degree in coexpression modules. We chose β = 10 based on the selected lncRNAs and mRNAs. Finally, we constructed a hierarchical clustering tree between genes with the Pearson coefficient. The different branches and colors of the clustering tree represent different gene modules. According to the Pearson coefficient, different modules were integrated into one module for further research [29].

### 2.3. Functional Enrichment Analysis

The biological process, molecular function, and cellular component information of the mRNA modules were analyzed based on the Gene Ontology (GO) database. Based on the gene signaling pathway annotation information of the Kyoto Encyclopedia of Genes and Genomes (KEGG) database, we performed the signaling pathway annotation of the mRNA in the modules. Fisher’s exact test and the multiple comparison test were performed to examine the significance level (*p* < 0.05 and false discovery rate < 0.05). Thus, the significant functions and signaling pathways of the mRNAs were identified.

### 2.4. Construction of lncRNA–mRNA Coexpression Networks and lncRNA–mRNA-Pathway Coexpression Network

Genes with higher connectivity in the module were named hub genes. After screening the hub genes, the weights of the coexpression relationship between the hub genes were calculated, and the lncRNA–mRNA and mRNA–mRNA coexpression regulatory relationships were selected to build the lncRNA–mRNA coexpression network. Then, the lncRNA–mRNA network and the significant signaling pathways regulated by mRNAs in module were used to construct the lncRNA–mRNA pathway network. The purpose of constructing the pathway network was to reveal the signaling pathways related to the regulation of lncRNAs to predict the potential mechanisms of lncRNAs in NAFLD.

### 2.5. Protein–Protein Interaction (PPI) Network Construction

The PPI network analysis was performed with an online tool (STRING; version 11.5; https://cn.string-db.org/ (accessed on 27 May 2022)) [30]. In the present study, all lncRNAs and mRNAs in the modules were analyzed with STRING.

### 2.6. In Vivo Experiment

The procedure for the animal experiments was approved by the Ethics Committee on the Use of Animals of Southeast University (protocol code 20220225025). We randomly assigned 30 C57BL/6 male mice (5 weeks old) to the control, HFD, or HFD + lycopene group. The HFD group was fed a HFD (60% kcal from fat; D12492; Nanjing-Xietong Inc., China); the HFD + lycopene group was fed a HFD mixed with lycopene (100 mg/kg·d); The control group was fed a normal diet (12% kcal% from fat). All mice were housed in a 12 h light/dark environment and had free access to food and water without fasting. They were sacrificed by cervical dislocation after 10 weeks of feeding, and liver tissues were immediately isolated. Each liver tissue was divided into two parts: one part was frozen in liquid nitrogen and stored at −80 °C; the other part was fixed with 4% paraformaldehyde.

### 2.7. Hematoxylin–Eosin Staining

The livers fixed with 4% paraformaldehyde were dehydrated, embedded in paraffin, sliced, stained with hematoxylin–eosin (HE), and observed under a microscope and quantified with multispectroscopy.

### 2.8. Statistical Analysis

Statistical analysis was performed with R software Version 4.1.1 (http://www.r-project.org/ (accessed on 1 February 2023)) and SPSS Version 21.0 (IBM Corp., Armonk, NY, USA). The results are presented as mean ± SD. Student’s *t*-test and one-way analysis of variance (ANOVA) were used to determine differences. *p* < 0.05 was considered significant.

## 3. Results

### 3.1. Weighted Coexpression Network Construction and Key Module Identification

A total of 1602 lncRNAs and 2202 mRNAs were identified from 9 samples from the GSE146627 dataset by cluster analysis (Appendix A). R language was used to verify the data integrity and construct a network topology to determine that the soft threshold power β value was 10 (Figure 2a). Then, based on the soft threshold, we constructed a coexpression matrix. Additionally, we constructed clustering dendrograms by calculating the gene adjacency and dissimilarity coefficients (Figure 2b). Eleven modules (turquoise, blue, brown, yellow, green, red, black, pink, magenta, purple, and grey) were separated with the dynamic shearing method, and the number of lncRNAs and mRNAs in these modules is shown in Table 1.

### 3.2. Correlation Analysis of Module and Traits

The relationship between the coexpression modules and dietary interventions is shown in Figure 3. We found that the turquoise module was most positively correlated with HFD + lycopene (correlation coefficient = 1, *p*-value = 2 × 10^−9^), while the black module was most positively correlated with HFD (correlation coefficient = 0.87, *p*-value = 0.002). Based on the findings in Table 1 and Figure 3, the lncRNAs and mRNAs in the turquoise, blue, brown, yellow, green, and black modules were selected for further network regulation analysis.

### 3.3. GO and KEGG Pathway Analysis

We performed enrichment analysis of the GO terms and KEGG pathways to determine whether these modules consist of functionally similar genes as well as to understand the functional significance of the network modules. The mRNAs in the turquoise, blue, brown, yellow, green, and black modules were subjected to GO and KEGG pathway enrichment analyses. The results showed that the regulation of transcription by RNA polymerase II, lipid metabolic process, apoptotic process, fatty acid metabolic process, steroid biosynthetic process, phosphorylation, RNA splicing, and positive regulation of I-kappa B kinase/NF-kappa B signaling were enriched in the top 20 GO terms, as shown in Figure 4a,b (turquoise and blue modules) and Appendix A (brown, yellow, green, and black modules). KEGG pathway enrichment analysis suggested that the differentially expressed mRNAs mainly played key roles in metabolic pathways, the TNF signaling pathway, and cytokine–cytokine receptor interaction (Figure 4c,d and Appendix A). Taken together, these results suggested a strong correlation between the genes in these modules and inflammation and metabolism.

### 3.4. Construction of lncRNA–mRNA Coexpression Networks

lncRNA–mRNA coexpression networks were built to detect the functional mechanisms of lncRNAs in the key modules (Figure 4). There were 46 lncRNAs and 38 mRNAs as hub genes in the turquoise module (Figure 5a). We detected 29 lncRNAs and 29 mRNAs as hub genes in the blue module (Figure 5b). Hub genes in the brown, yellow, green, and black modules are shown in Appendix A. We found that lncRNAs and mRNAs were inter-regulated. Moreover, one lncRNA was coexpressed with multiple mRNAs, and multiple lncRNAs were coexpressed with one mRNA, which revealed a comprehensive regulatory association in the lncRNA–mRNA coexpression networks.

### 3.5. Construction of lncRNA–mRNA Pathway Network

To identify the potential mechanisms through which the lncRNA-mediated regulation of signaling pathways occurs, we interacted the significantly different pathways and the lncRNA–mRNA coexpression network to obtain the lncRNA–mRNA pathway network (Figure 5 and Appendix A). In the pathway network, the turquoise module (Figure 5c) had 20 lncRNAs and 7 mRNAs, and the blue module (Figure 5d) had 12 lncRNAs and 4 mRNAs. In the turquoise module, Smarca2 (a lncRNA) was linked to three mRNAs (Cir1, Ralbp1, and Lpin1) and enriched in Epstein–Barr virus infection, metabolic pathways, mTOR signaling pathway, glycerophospholipid metabolism, glycerolipid metabolism, pathways in cancer, pancreatic cancer, and Ras signaling pathway. Tacc1 (a lncRNA) was related to 3 mRNAs (Prkce, Ube3a, and Efna5) and abundant in 19 signaling pathway, such as Axon guidance, PI3K–Akt signaling pathway, MAPK signaling pathway, Rap1 signaling pathway, microRNAs in cancer, Ras signaling pathway, insulin resistance, type II diabetes mellitus, and human papillomavirus infection.

In the blue module, Flywch1 (a lncRNA) was linked to three mRNAs (Foxp3, Mef2c, and Serpine1) and enriched in inflammatory bowel disease; transcriptional misregulation in cancer; parathyroid hormone synthesis, secretion, and action; oxytocin signaling pathway; cGMP–PKG signaling pathway; MAPK signaling pathway; Apelin signaling pathway; fluid shear stress and atherosclerosis; AGE–RAGE signaling pathway in diabetic complications; HIF-1 signaling pathway; p53 signaling pathway; and cellular senescence. Rnf169 (a lncRNA) was also linked to two mRNAs (Foxp3 and Mef2c) and abundant in inflammatory bowel disease; transcriptional misregulation in cancer; parathyroid hormone synthesis, secretion, and action; oxytocin signaling pathway; cGMP–PKG signaling pathway; MAPK signaling pathway; Apelin signaling pathway; fluid shear stress; and atherosclerosis.

### 3.6. Construction of Protein–Protein Interaction Network

To investigate the interactive relationships among the hub genes in the modules, we submitted the hub genes to the STRING database to construct PPI networks (Figure 6 and Appendix A). The PPI network in the turquoise module is shown in Figure 6a; there are 28 key genes in the turquoise module according to PPI analysis. The PPI network in the purple module is shown in Figure 6b; there are 23 key genes in the blue module according to PPI analysis.

### 3.7. Hematoxylin–Eosin Staining

After 10 weeks of normal diet, HFD, and HFD + lycopene feeding, liver tissues were detected with HE staining. The result showed that compared with the normal-diet group (Figure 7a), the hepatocytes in the HFD group appeared swollen and loosely arranged, and various sizes and numbers of lipid droplet vacuoles were found in most hepatocytes (Figure 7b). This suggested that HFD could cause hepatocyte steatosis in mice, thus promoting the development of NAFLD. However, HFD + lycopene feeding resulted in reductions in hepatocyte swelling, hepatocyte steatosis, and large fat vacuoles compared with those of the HFD group (Figure 7c). The result indicated that lycopene supply attenuated hepatic steatosis in HFD-induced mice.

## 4. Discussion

In recent years, a number of genes regulating NAFLD have been identified based on genomic studies [31,32,33]. Most studies have detected gene expression differences between NAFLD patients and heath controls [34,35,36]. However, the pathogenesis of NAFLD was still unclear. A recent study showed an intimate relationship between NAFLD and periodontitis, which was utilized for the acquisition of modules and the construction of a miRNAs–mRNAs network up with NAFLD and periodontitis with WCGNA [37]. In the present study, we investigated the interaction between a HFD and lycopene on NAFLD. A total of 1602 lncRNAs and 2202 mRNAs were identified based on the GEO database (GSE146627 dataset). After WGCNA, we divided the lncRNAs and mRNAs into 11 modules with various functions, and found 6 modules (turquoise, blue, brown, yellow, green, and black modules) that were closely related to NAFLD. We used functional enrichment analysis to explore the potential biological effects of lncRNAs and mRNAs in NAFLD. In contrast with previous studies on the relationship between mRNA and NAFLD [23,24], the lncRNA–mRNA co-expression network presented a novel insight into the relationship between lncRNA and mRNA. Moreover, we constructed a lncRNA–mRNA pathway coexpression network and found a positive function on the MAPK signaling pathway, HIF-1 signaling pathway, p53 signaling pathway, Epstein–Barr virus infection, metabolic pathways, mTOR signaling pathway, glycerophospholipid metabolism, glycerolipid metabolism, pathways in cancer, and Ras signaling pathway, among other signaling pathways. This indicated that NAFLD is caused by metabolic disorders induced by multiple signal pathways and revealed the regulatory role of lycopene in NAFLD.

Compared with the differentially expressed genes, WGCNA used the expression information of thousands of genes to identify gene modules and perform association analysis with phenotypes [38]. The hub genes in WGCNA play key roles in the regulation of biological functions [39,40]. Hence, exploring the hub genes in the modules is important for understanding the development of NAFLD. Moreover, we focused on the hub genes in the turquoise, blue, brown, yellow, green, and black modules. In these six modules, we found that many hub mRNAs had positive effects on metabolism and inflammation. This suggested that metabolic disorders and immune dysregulation induced by multiple genes cause the development of NAFLD [41,42].

Increasing numbers of studies are suggesting that lncRNAs play an important role in the development and progression of various types of diseases [12,43]. In the turquoise module, Smarca2, Tacc1, Calcoco1, Cln3, BB365896, and Gm38366 were key lncRNAs linked to hub mRNAs (Cir1, Lpin1, Ralbp1, Efna5, Prkce, Ube3a, and Nfkbia). NFKB inhibitor alpha (*Nfkbia*), a key gene in the NF-κB signaling pathway, was expressed in elevated levels in HepG2 cells during steatosis and inflammation [44]. Silencing lncRNA lncTNF suppressed the expression of Nfkbia and reduced NF-κB activity. Ralbp1 might function as a proinflammation molecule, and downregulated Ralbp1 significantly inhibited lipopolysaccharide-induced inflammation and liver damage in sepsis [45,46]. Zuniga et al. [47] proved that Prkce synergistically interacts with TLR4 to promote NF-kB activation. Therefore, we speculated that BB365896, Smarca2, Calcoco1, and Tacc1, which were coexpressed with Nfkbia, Ralbp1, and Prkce in our WGCNA, might play an inflammation-modulating role during NAFLD. Clifford et al. [48] reported that Lpin1 reduced the hepatic levels of monounsaturated fatty acids to treat NAFLD. In addition, another study showed that Lpin1 might play an important part in hepatocyte lipid metabolism [49]. MLL4 was defined as a critical regulator of overnutrition that induced obesity and hepatic steatosis [50]. Moreover, Ube3a suppressed overnutrition-induced NAFLD by targeting and degrading MLL4 [51]. Calcoco1, Smarca2, Cln3, Tacc1, and Gm38366 were coexpressed with Lpin1 and Ube3a. Therefore, we speculated that BB365896, Calcoco1, Smarca2, Cln3, Tacc1, and Gm38366 might be involved in the progression of NAFLD by regulating the Ras signaling pathway, aldosterone synthesis and secretion, AGE–RAGE signaling pathway in diabetic complications, cGMP–PKG signaling pathway, sphingolipid signaling pathway, inflammatory mediator regulation of TRP channels, insulin resistance, TNF signaling pathway, cAMP signaling pathway, adipocytokine signaling pathway, metabolic pathways, mTOR signaling pathway, and glycerolipid metabolism.

In the blue module, Flywch1 was the core lncRNA connected to three hub genes (Foxp3, Mef2c, and Serpine1). Foxp3 has been reported to be involved in the development of NAFLD through the regulation of lipid metabolism and the inflammatory microenvironment [52,53]. Additionally, Mef2c has been connected with inflammatory diseases in humans and animals [54,55]. Recent research showed that adipose tissue overexpressed Serpine1 in NAFLD, which generally induced inflammation [56]. Moreover, injected MERTK+/hi M2c macrophages inhibited NAFLD progression by suppressing SERPINE1 [57]. Flywch1 was found to be a master regulatory gene in coronary artery disease and affected cholesterol-ester accumulation in foam cells [58]. We predict that Flywch1 might interact with Foxp3, Mef2c, and Serpine1 and induce NAFLD progression through inflammatory bowel disease, cGMP–PKG signaling pathway, MAPK signaling pathway, fluid shear stress and atherosclerosis, AGE–RAGE signaling pathway in diabetic complications, HIF-1 signaling pathway, and p53 signaling pathway.

In our present study, we found that feeding lycopene attenuated hepatic steatosis in HFD-induced NAFLD in mice. Lycopene was shown to inhibit the MAPK and NF-κB pathways to reduce superoxide synthesis [59]. Gouranton et al. [60] demonstrated that lycopene treatment limited proinflammatory cytokine and chemokine production through the NF-κB pathway in adipose tissue in mice. Similar results were found in a recent study: supplementation of lycopene may have beneficial effects on hepatocyte metabolism to prevent and treat nonalcoholic hepatic steatosis [61].

## 5. Conclusions

Based on WGCNA, we identified NAFLD-related candidate hub lncRNAs and mRNAs, and we constructed lncRNA–mRNA coexpression networks and lncRNA–mRNA pathway coexpression networks. Our analysis revealed that the hub lncRNAs may regulate the inflammatory and metabolic pathways that are crucially involved in the pathogenesis of NAFLD. Moreover, lycopene has the potential to be a phytochemical for the prevention of HFD-induced liver steatosis. Therefore, in the future, researchers should investigate the regulatory role between lycopene and key lncRNAs on NAFLD.

## Figures and Tables

**Figure 1 nutrients-15-00948-f001:**
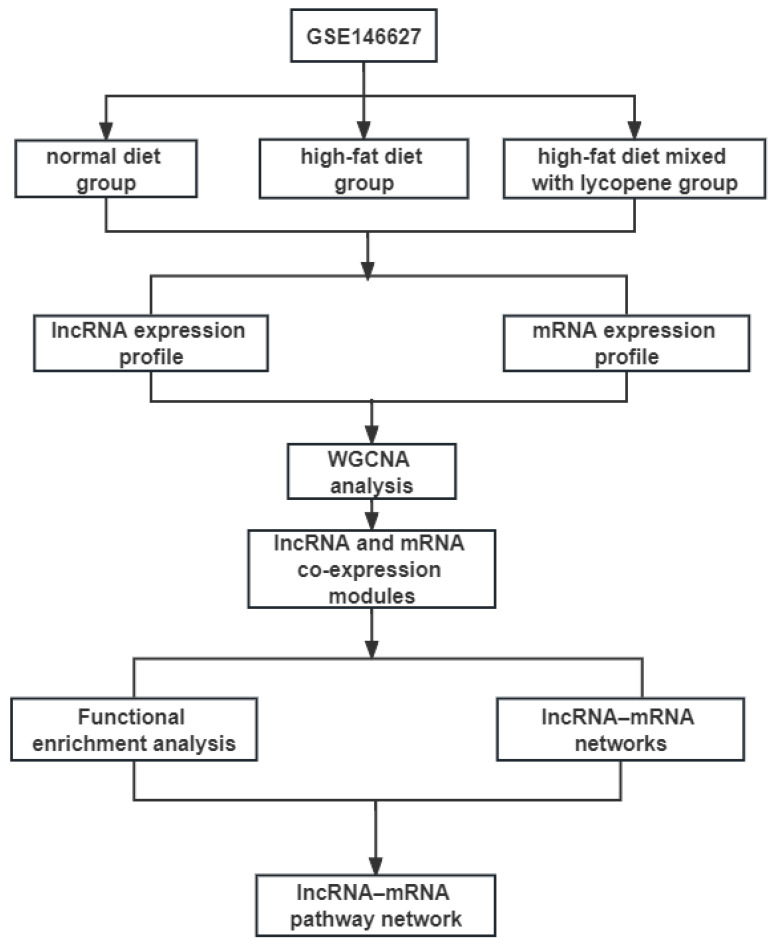
The main workflow of the study. WGCNA, weighted correlation network analysis.

**Figure 2 nutrients-15-00948-f002:**
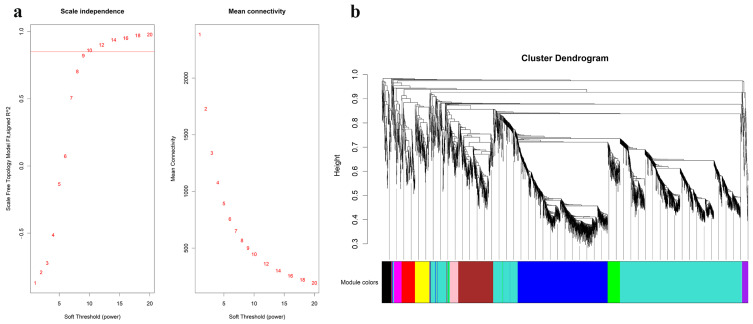
WGCNA was used to identify trait-related modules and genes. (**a**) Scale independence and mean connectivity analysis for various soft threshold powers. (**b**) Clustering dendrograms of lncRNAs and mRNAs. The color bands provide a simple visual comparison of module assignments based on the dynamic tree cutting method. WGCNA, weighted correlation network analysis.

**Figure 3 nutrients-15-00948-f003:**
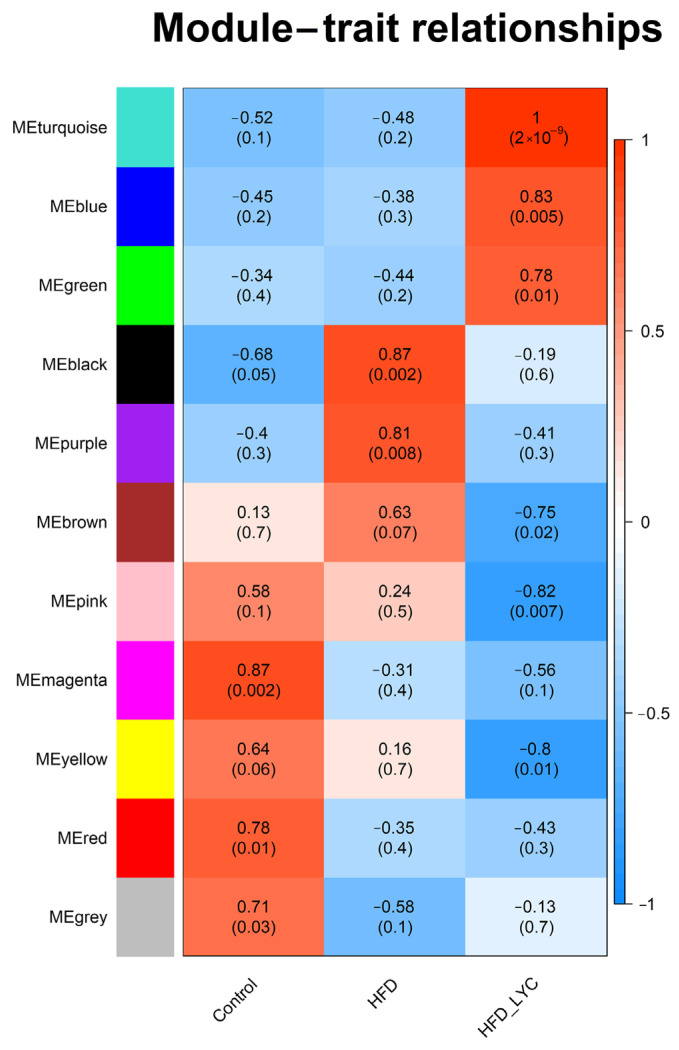
Module–trait relationship. Each row represents a module eigengene and each column represents a trait. Each cell includes the corresponding correlation and *p*-value. HFD, high-fat diet; HFD-LYC, high-fat diet + lycopene.

**Figure 4 nutrients-15-00948-f004:**
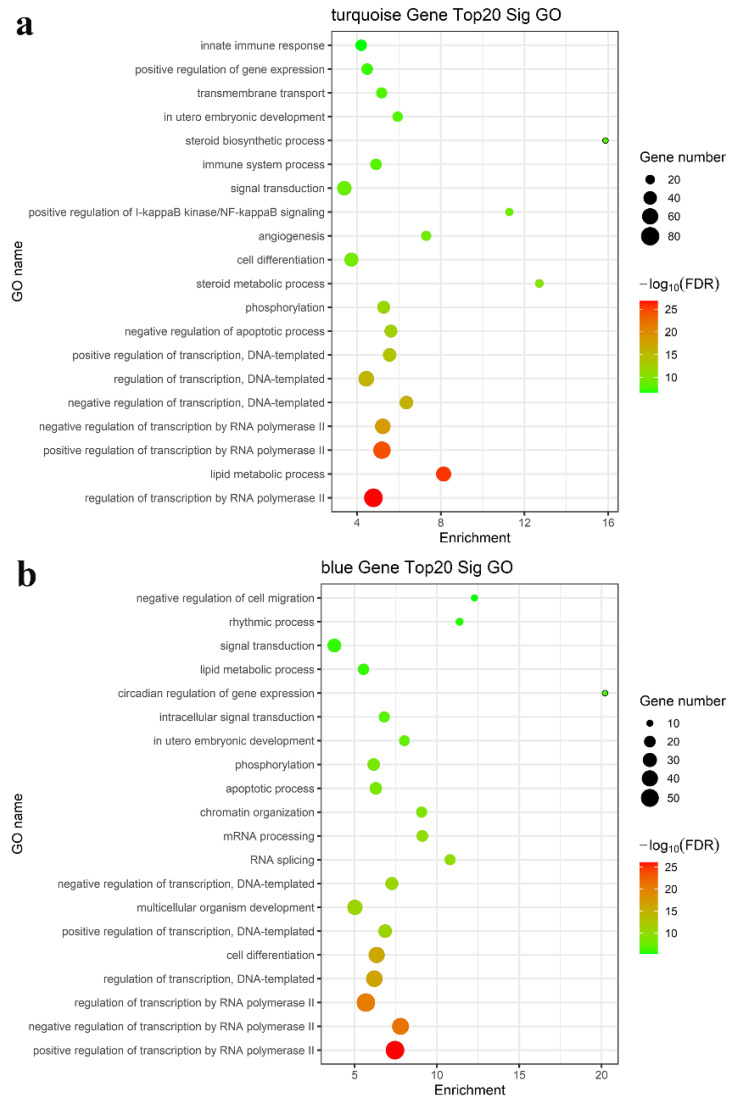
Functional enrichment analysis for the turquoise and blue modules. Enriched GO analysis of mRNAs in the (**a**) turquoise and (**b**) blue modules; enriched KEGG pathway analysis of mRNAs in the (**c**) turquoise module and (**d**) blue modules.

**Figure 5 nutrients-15-00948-f005:**
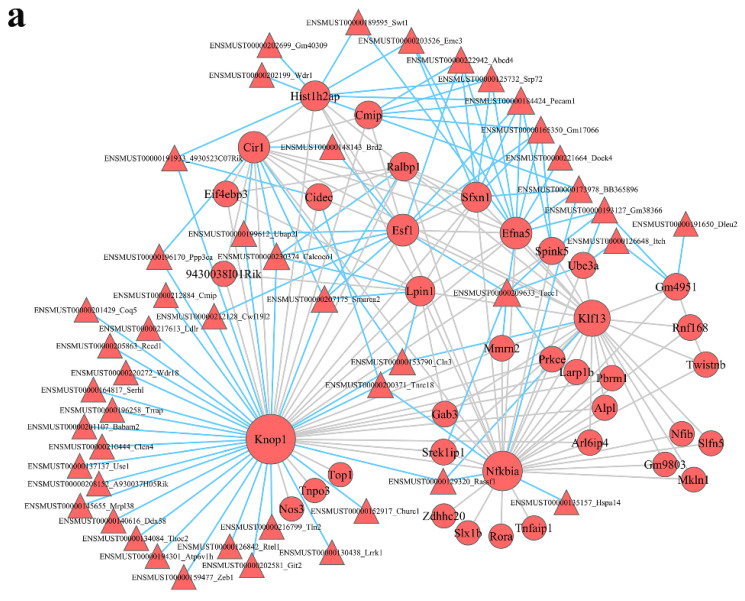
Module lncRNA–mRNA network of hub genes in the turquoise (**a**) and blue (**b**) modules. Module lncRNA–mRNA pathway network of hub genes in the turquoise (**c**) and blue (**d**) modules. Red circles represent mRNAs, red and yellow triangles represent lncRNAs, and gray polygons represent pathways. The size of the circle represents the regulation capacity of the mRNA, where larger circles indicate relatively stronger regulation capacity.

**Figure 6 nutrients-15-00948-f006:**
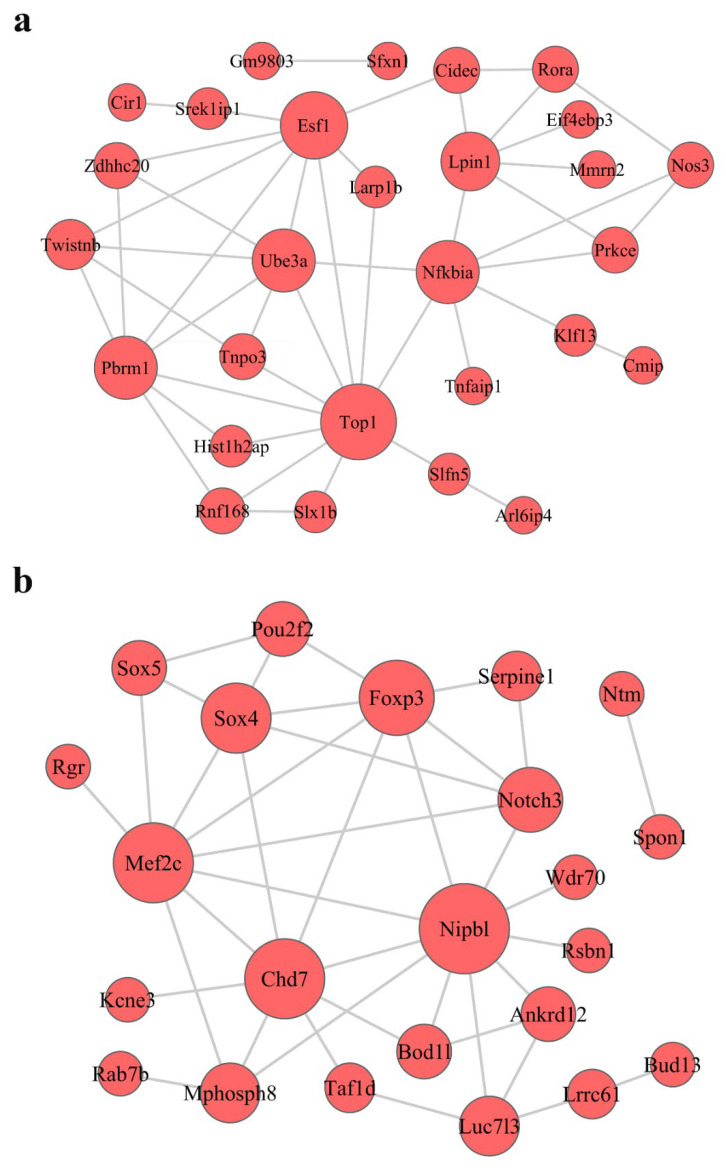
Construction of protein–protein interaction (PPI) networks in the turquoise (**a**) and blue (**b**) modules. Red circles represent mRNA; the size of the circle represents the regulation capacity of mRNA, where larger circles indicate relatively stronger regulation capacity.

**Figure 7 nutrients-15-00948-f007:**
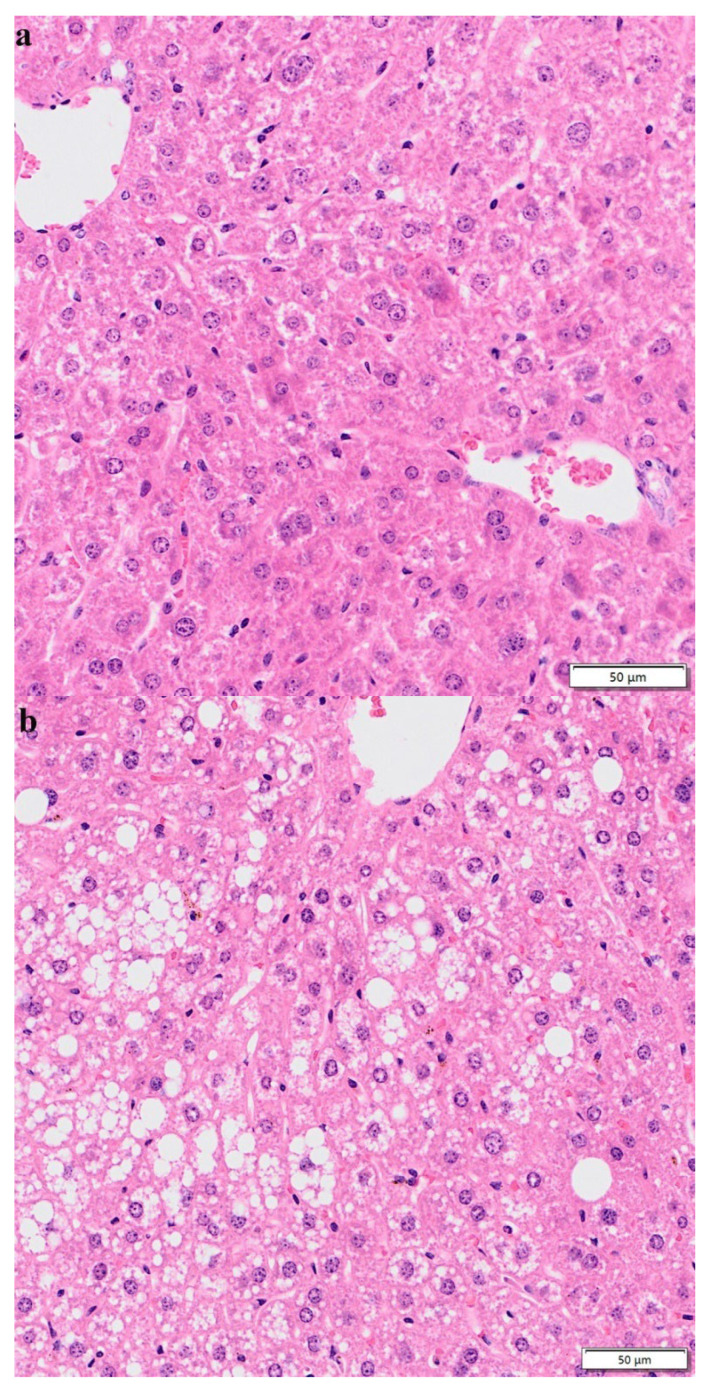
HE staining of the liver sections: (**a**) control group; (**b**) high-fat diet group; (**c**) high-fat diet + lycopene group. Magnification 200×.

**Table 1 nutrients-15-00948-t001:** The number of lncRNAs and mRNAs in the 11 modules.

Module	Total	No. of lncRNAs	No. of mRNAs
turquoise	1700	834	860
blue	955	482	468
brown	370	31	339
yellow	152	40	112
green	144	8	136
red	142	60	82
black	101	31	70
pink	92	5	87
magenta	83	51	32
purple	63	49	14
grey	2	2	

## Data Availability

Data available in a publicly accessible repository.

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
