# Peer review of "Identification and Evaluation of Hub Long Noncoding RNAs and mRNAs in High Fat Diet Induced Liver Steatosis"

_nutrients, 2023, doi:10.3390/nu15040948_

Round 1

Reviewer 1 Report

Abstract page 1 lines 21 - 23 - This is not a finding of this paper, you assume it.

The results section - page 5 figure 1 - is too small to be legible to any reader.

The results section - page 7 section 3.3 - you need to make this clearer, so what is written is a clear reflection of exactly what your study did and found. There should be no suggestion of what the results mean. You're not pitching the results.  

The results section - page 8 figure 4 - is too small to be legible to any reader.

The results section - page 9 figure 5 - is too small to be legible to any reader.

The results section - page 9 lines 200-203; lines 207-209; lines 216-218; page 10 lines 219-220 - in all of these sections you 'discuss' other people's work (without citing) as if it's your own. Change this! remove reference to what results mean unless it is in the discussion! 

The results section - page 5 figure 1 - is too small to be legible to any reader. I am assuming given they are red circles that they are mRNA PPI? It is not clear and given the confusing legend for figure 5, more assumptions need to be made by the reader. 

The results section - page 10 lines 236-238 - You did not determine increased swelling! In the methods section 2.7 you describe staining and taking pictures. So is this inflammation determined via sight? The pictures are too small how did you determine a 30% increase? This is not transparent. 

The Discussion section - page 12 lines 296-298 - These sentences are not formatted correctly. You 'discuss what others have found in their research' and then related to your findings. You simply state a finding as if the work is directly related to your results. 

-The Discussion section - page 12 line 300 'A last year's research'? This does not make sense.

The Discussion section - page 12 lines 309-310 You did not determine that Lycopene supply attenuated hepatic steatosis in HFD-induced NAFLD! This is misleading, keep your findings realistic. 

The conclusion - This should be redone to discuss your findings only.

The last two sentences are repeated. 

Reviewer 2 Report

Line 92: "Any specific reason for choosing beta 10?"

Line 99: Suggestion to include why the module is selected.

Line 244–260: This is a repeat of the results, did not compare with any previous study, and stated the reason for NAFLD. Comparing it with the literature and discussing the reason for the NAFLD will be better.

Line 322: It will be helpful to the readers if you can include future studies.

Round 2

Reviewer 1 Report

Thank you for making Changes and addressing concerns.

The minor changes required to publish this paper are again the size of the images. Although they have increased I still need to enhance the size of the document to even see the font. The font in each image should be as legible as the font in the text. The font size in Figure 3 is just ok - no smaller. 

Figure S2 for example should have 4 images on a page, not 8.

Figures 1, 4 (a, b, c, d) and figure 5 and 6 all need to be larger and separated on the page. 

Figure 7 is now clearer but grossly oversized compared to the other images. 

There needs to be consistency in the formatting. 
